# Direct imaging of light-element impurities in graphene reveals triple-coordinated oxygen

Christoph Hofer [1,2,3]*, Viera Skákalová [1], Tobias Görlich[1], Mukesh Tripathi[1], Andreas Mittelberger [1], Clemens Mangler[1], Mohammad Reza Ahmadpour Monazam[1], Toma Susi [1], Jani Kotakoski [1] & Jannik C. Meyer [1,2,3]

Along with hydrogen, carbon, nitrogen and oxygen are the arguably most important elements for organic chemistry. Due to their rich variety of possible bonding configurations, they can form a staggering number of compounds. Here, we present a detailed analysis of nitrogen and oxygen bonding configurations in a defective carbon (graphene) lattice. Using aberration-corrected scanning transmission electron microscopy and single-atom electron energy loss spectroscopy, we directly imaged oxygen atoms in graphene oxide, as well as nitrogen atoms implanted into graphene. The collected data allows us to compare nitrogen and oxygen bonding configurations, showing clear differences between the two elements. As expected, nitrogen forms either two or three bonds with neighboring carbon atoms, with three bonds being the preferred configuration. Oxygen, by contrast, tends to bind with only two carbon atoms. Remarkably, however, triple-coordinated oxygen with three carbon neighbors is also observed, a configuration that is exceedingly rare in organic compounds.

[1] Faculty of Physics, University of Vienna, Boltzmanngasse 5, A-1090 Vienna, Austria. [2] Institute for Applied Physics, Eberhard Karls University of Tuebingen, Auf der Morgenstelle 10, D-72076 Tuebingen, Germany. [3] Natural and Medical Sciences Institute at the University of Tuebingen, Markwiesenstr. 55, D-72770 Reutlingen, Germany. *email: christoph.hofer@uni-tuebingen.de

Recent advances in transmission electron microscopy, in particular aberration correction, have enabled the study of low-dimensional materials at low electron energies with atomic resolution. In scanning transmission electron microscopy (STEM)[1], the contrast mechanism behind annular dark field images allows the identification of light elements (e.g. B, C, N, O) despite their very similar atomic number[2]. In aberration-corrected high-resolution transmission electron microscopy (HRTEM), however, these elements have an almost-identical contrast and their discrimination becomes difficult in particular when they are incorporated into irregular structures such as defects[3,4]. Atomic resolution images have revealed the bonding configurations of several types of impurities in light-element samples. For example, nitrogen dopants in graphene and carbon nanotubes have been revealed in several studies[5–9], boron dopants have been identified in graphene by STEM[8], and carbon and oxygen impurities have been revealed in monolayer hexagonal boron nitride[2]. Oxygen impurities in graphene are of high relevance due to their importance for the processing of graphene oxide (GO), and are likely to play a role, e.g. in the degradation of graphene in oxygen or in air at high temperatures. Despite efforts to quantify the functional groups in GO[10–15], the nature of oxygen binding to graphene is still not well understood. Although few HRTEM studies have revealed disorder and defects in graphene oxide[16–19], a direct visualization of oxygen atoms that includes their unambiguous chemical identification (e.g., via contrast in STEM or via electron energy loss spectroscopy, EELS) along with their bonding with a carbon matrix has not been achieved yet.

Here, we study a large number of oxygen impurities in samples of graphene oxide. The oxygen atoms are identified by their contrast in medium-angle annular dark field STEM images[2], and in several cases also by EELS. For comparison, we also prepared a graphene sample with nitrogen impurities by low-energy plasma and ion treatment (see the "Methods" section). In contrast to an earlier study[8], our samples are transferred under vacuum from implantation to STEM imaging, which prevents configurations with open bonds from being saturated with contamination. Our data set is large enough to carry out a statistical analysis of the different bonding configurations for oxygen and nitrogen. Moreover, we describe the dynamics of reduction observed under the electron beam for the case of oxygen.

## Results

**Configurations**. Before discussing the observed atomic configurations, it must be pointed out that initial changes occur in the structure of graphene oxide already at relatively low doses, which makes it challenging if not impossible to capture the pristine structure in atomic resolution TEM or STEM images. In agreement with earlier findings[20], we observed changes in the EELS signal at doses between $10^3$ and $10^6$ e$^-$ Å$^{-2}$ (Fig. 1a–d). We assume that functional groups which are attached to the basal plane, edges, or defects of graphene via relatively weak bonds (such as hydroxyl, carboxyl, epoxide, or ketone groups) are destroyed at these doses before an image could be obtained. Nevertheless, what remains after initial electron irradiation is a defective graphene sample, where numerous oxygen impurities are incorporated into a carbonaceous host structure. These structures, which are stable enough for STEM imaging and in some cases EELS, reveal a variety of bonding configurations for oxygen in an $sp^2$-bonded carbon system.

A high-magnification image where the defective carbon honeycomb lattice can be resolved is presented in Fig. 1e. While the regular graphene lattice dominates the area of the sample, a remarkably high density of defects with brighter impurity atoms can be identified. By analyzing the intensities[2], most of these atoms can be assigned as oxygen. Figure 1f, g show the histogram of the intensities. We further confirmed the identity for some of the impurities by EELS (which in turn validates the intensity analysis). Due to a lower dose than that used in ref. [2] necessitated by the sample stability, a small fraction of the impurities cannot be uniquely assigned, e.g. where the tails between the nitrogen and oxygen intensity distribution overlap in the histogram (Fig. 1g). For example, the atom in Fig. 1c marked by the blue circle could—based on the intensity alone—be either nitrogen or oxygen. However, since the EELS signal of the GO samples shows no indication of nitrogen, we assume the impurity atom to be oxygen in such cases.

For the N-doped graphene, we classify the configurations in agreement with earlier literature into graphitic (substitution with three N–C bonds), pyridinic (two N–C bonds in a hexagon) and pyrrolic (two N–C bonds in a pentagon) configurations. Different oxygen and nitrogen configurations are shown in Fig. 2.

For the oxygen impurities, the conventional classification into different types of functional groups is not useful for describing the observed structures. Instead, we classify the oxygen configurations into three frequently observed types. The first, and surprisingly frequently observed, configuration consists of two oxygen atoms substituting two neighboring carbon atoms. An example of a STEM image of this configuration is shown in Fig. 2a. Graphitic substitutions are our second type of configuration (Fig. 2b). This is the only configuration in which oxygen binds with three carbon neighbors, similar to the oxygen impurities imaged in hexagonal boron nitride[2]. Our third class of configurations are oxygen atoms next to vacancies. Interestingly, they form defect reconstructions that are very similar to those in graphene without heteroatoms, except that one or two carbon atoms at the edge of a vacancy are replaced by oxygen. We label these defects in accordance with the carbon-only structures. In a 5–9 monovacancy (MV)[21], for example, a single oxygen replaces the carbon atom with only two bonds (Fig. 2c, first column) while the structure undergoes a distortion that looks like the Jahn–Teller distortion of a carbon-only vacancy. If two oxygen atoms are present, however, both remain two-coordinated and the bond at the pentagon remains open leading to a symmetric MV configuration (Fig. 2c, second column). Another prominent example is the divacancy (DV)[22], where two oxygen atoms can sit in the same pentagon of a 5-8-5 DV (third column of Fig. 2c). Also here, the two oxygen atoms do not form a bond and have a larger projected distance than the corresponding carbon atoms at the opposite pentagon. The 555–777 DV (fourth column) shows an interesting behavior when one carbon is replaced by an oxygen atom: Here, the oxygen only binds with two carbon atoms, breaking the three-fold symmetry. In a configuration where three carbon atoms are missing (last column of Fig. 2c), the oxygen atom binds with two carbon atoms building a "bridge". This appears very similar to a graphene trivacancy (TV). All of these configurations, except for the graphitic type, form an ether-like bond, i.e. an oxygen binding to two different carbon atoms.

A statistical analysis of the distribution of the configurations reveals that the oxygen pair is the most prominent one (Fig. 2e), whereas the graphitic substitution is the least frequent. In contrast, in our N-doped graphene sample, the pyrrolic configuration was the most favored one followed by the graphitic substitution. The occurrence of the pyridinic configuration is low in our case. It can be increased by ozone treatment during sample preparation[23], which we have not done. The difference of the bonding configurations of N and O in graphene can be highlighted by the distribution of their coordination numbers (Fig. 2f). The statistical analysis of our atomic resolution images

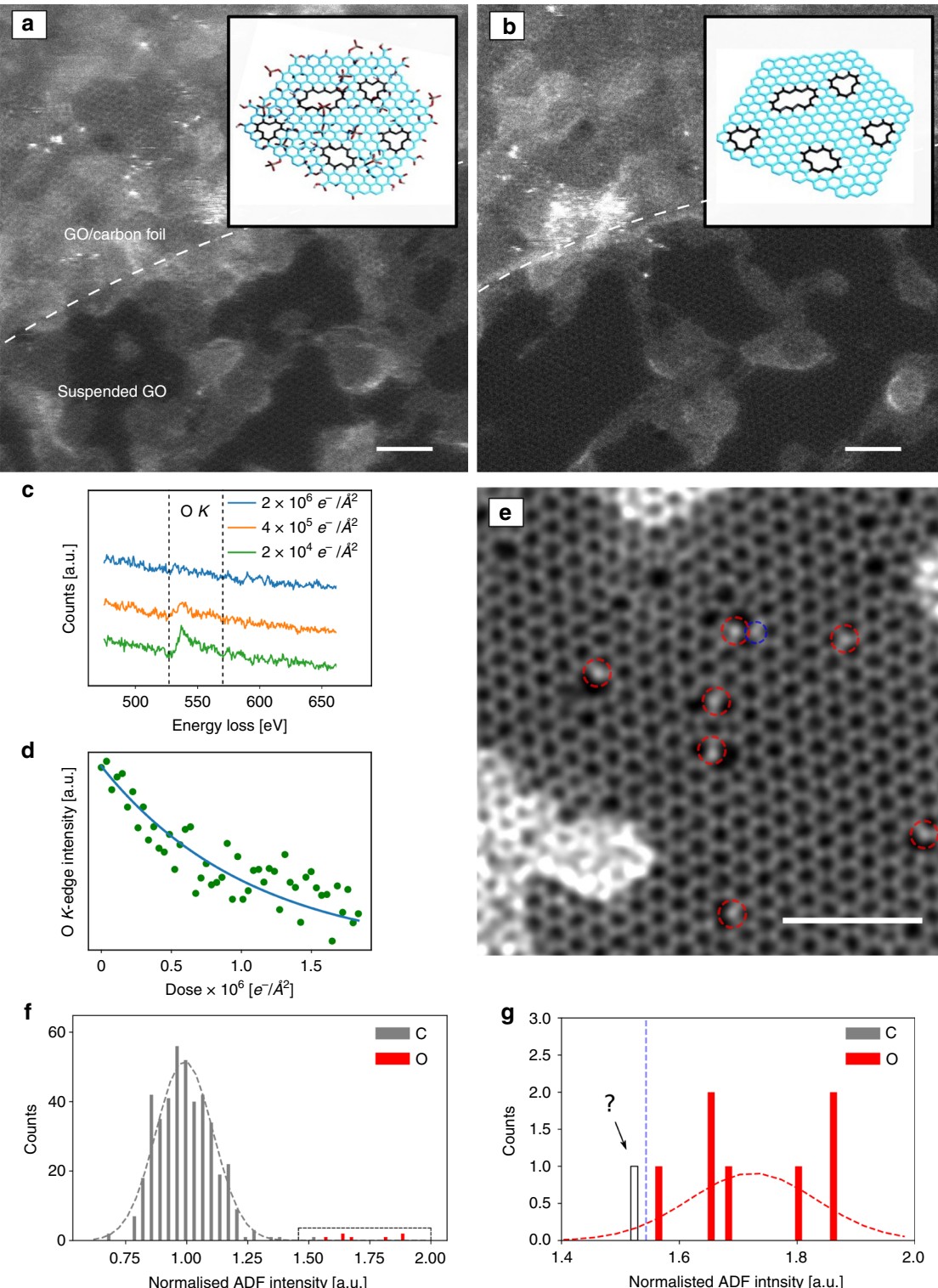

**Fig. 1** Reduction of GO under the beam. Lower magnification (lattice resolution) STEM images: **a** initial and **b** after ~50 scans. Adsorbates shrink under observation, and the clean lattice area increases. The insets show a model with and without functional groups attached to the graphene sheet (reduced GO). The upper left section of the image contains the supporting carbon film. **c** EEL spectra after different electron doses showing the loss of the oxygen *K*-edge. **d** EEL intensity of the oxygen *K*-edge as a function of electron dose. **e** High magnification double-Gaussian filtered image where the graphene lattice with defects and impurities is resolved. The bright atoms (red dashed circles) can be identified as oxygen. The atom in the blue dashed circle is at the edge of the intensity distribution and might be either nitrogen or oxygen. **f** Histogram of the ADF intensities of carbon (gray) and oxygen atoms (red). **g** Magnified histogram of panel **f**. Insets in **a**, **b** are reprinted from ref.[10] with permission. Scale bars are 2 nm

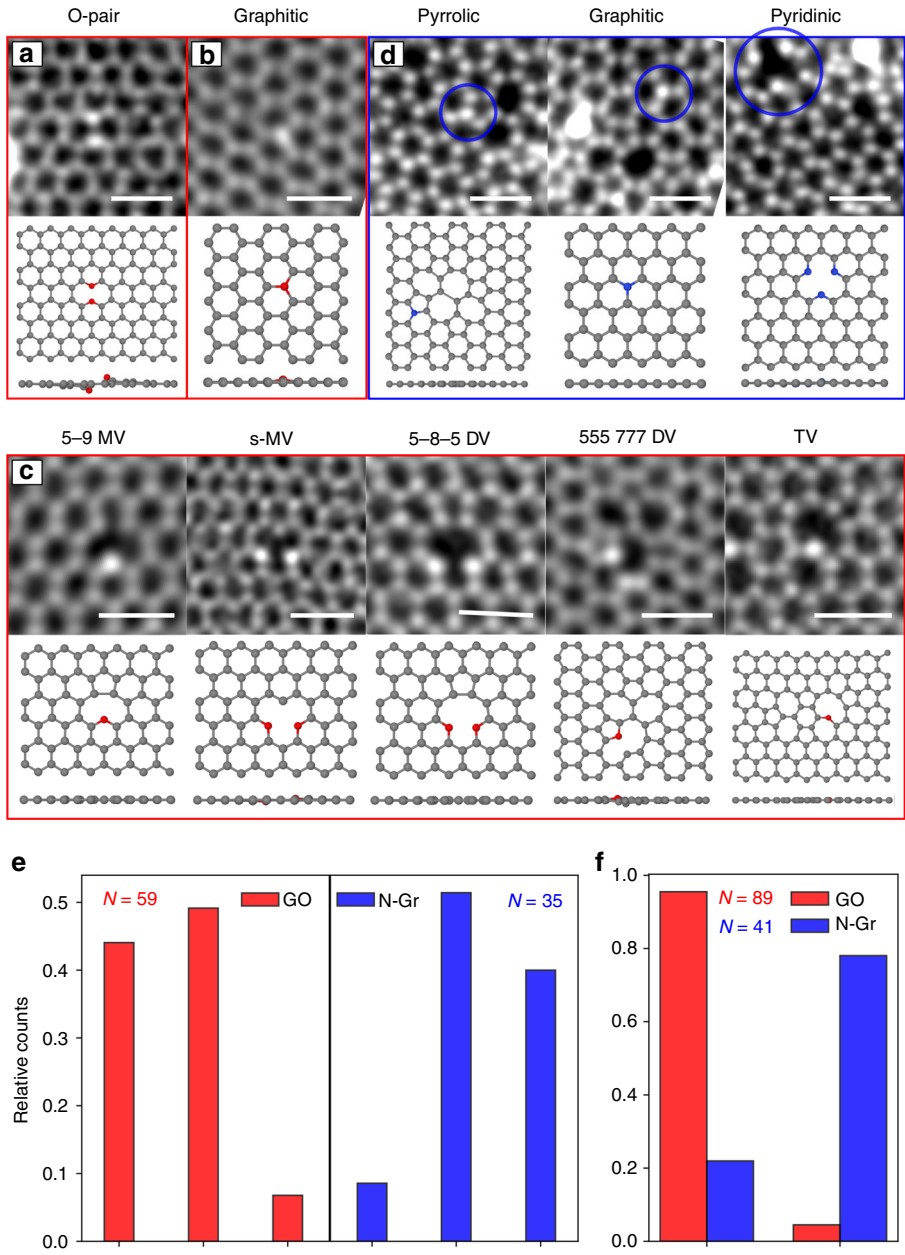

**Fig. 2** STEM images of different configurations of oxygen and nitrogen atoms in graphene. **a** Oxygen pair. **b** Graphitic substitution by oxygen. **c** Oxygen atoms within vacancies. **d** Nitrogen-doped graphene configurations. **e** Distribution of the different configurations in GO (red) and N-doped graphene (blue). **f** Distribution of double-coordinated and triple-coordinated heteroatoms in GO (red) and N-doped graphene (blue). $N$ shows the total number of heteroatoms of each sample (note that some configurations contain multiple heteroatoms). Scale bars are 0.5 nm

directly confirms that oxygen prefers two bonds while nitrogen prefers three bonds. This is in agreement with the different electronic configuration of these elements, and hence different preferences for forming chemical bonds with carbon.

To analyze their structural properties, we performed density functional theory (DFT) calculations for each configuration. The obtained relaxed models are shown below the STEM images in Fig. 2. In some cases, we find clear differences in the structural relaxation for oxygen in comparison to nitrogen. For example, the relaxed structure of the 555–777 DV (Fig. 2c, fourth column) shows a large projected distance of the oxygen atom to one of its three neighbors, breaking the three-fold symmetry of this configuration. Clearly, the oxygen in this case binds only with

two carbon atoms. However, a nitrogen atom in the same position results in a highly symmetric configuration with three neighbors close to the impurity (Supplementary Fig. 2). For the double-oxygen site (Fig. 2a), the two oxygen atoms do not bind but stick out of the graphene plane in opposite directions, with a projected distance that is significantly larger than the carbon–carbon bond in graphene. This is not the case for a simulated double-nitrogen structure (see Supplementary Discussion). All other considered configurations have very similar structural properties for both nitrogen and oxygen impurities, except for small out-of-plane displacements.

The energies of the relaxed configurations with the heteroatoms incorporated in the lattice ($E_{in}$) for both, nitrogen and

oxygen, were also obtained by DFT. We calculated the sum of the energies ($E_{out} + E_{isolated}$) when the heteroatoms are released from the lattice (after relaxation) and the isolated atoms (half of $N_2$ or $O_2$). The difference $E_{in} - (E_{out} + E_{isolated})$ is referred as binding energy and is lower (meaning higher stability) for all N configurations. This is in agreement with the observed higher stability of N dopants in graphene. All calculated energies are listed in Supplementary Table 2. The binding energies are negative in all cases, which means that the structures are stable with respect to forming a carbon-only vacancy plus isolated O or N.

**Dynamics.** The oxygen substitutions are sputtered after a dose with a geometrical mean of $5 \times 10^5 \, e^- \, Å^{-2}$ and replaced by a carbon atom, whereas nitrogen substitutions can withstand orders of magnitude higher doses[24] (cf. Fig. 3a–c). To understand this difference, we performed DFT-based molecular dynamics calculations (see the "Methods" section). The threshold energy for removing an oxygen atom from the lattice is 10.3 eV, and the threshold for removing the neighboring carbon is 15.0 eV. This energy is almost 2/3 of the calculated 22.0 eV threshold kinetic energy for a carbon in pristine graphene and also significantly lower than for nitrogen in graphene (19.09 eV)[25]. Indeed, graphitic nitrogen in graphene was found to be extremely stable

under 60 or 80 kV electron irradiation, such that the atomic structure of the dopant site is more likely to be changed by displacing a neighboring carbon atom before the dopant atom itself is sputtered under electron irradiation[25]. Hence, the observed clear difference in the stability under the beam between graphitic oxygen and nitrogen impurities is in agreement with the calculations. We further performed intensity analysis of another STEM image, where multiple oxygen pair configurations, as well as a graphitic configuration are present (see Fig. 3d). The histogram of the atom intensities (Fig. 3e) shows that the intensity of the graphitic configuration is clearly within the distribution of oxygen.

As mentioned above, we often observed the neighboring double-oxygen configuration as shown in Fig. 4a. Calculations[26] and experiments[27] show that directly neighboring nitrogen atoms are energetically unfavorable, while our experiments indicate that in the case of oxygen such a configuration is stable. Interesting dynamics can be observed when one oxygen atom is sputtered and a MV configuration with a single oxygen atom is left behind (Fig. 4b, c): During imaging, the oxygen atom jumps frequently to the opposite vacancy site. Similar dynamics were reported in a N-doped sample[5]. The number of images between such events spans the range from 1 to 15 with a dose of ca. $6 \times 10^5 \, e^- \, Å^{-2}$ per image. After a long electron exposure, the second O atom can be removed, leaving behind a DV (Fig. 4d), which is also highly dynamic[22]. A video of this process is shown in the Supplementary Discussion.

Another example of dynamics is shown in Fig. 4f. In this case, two oxygen atoms were found in a MV. After a few images, one carbon is sputtered and a 5-8-5 DV with two oxygen atoms forms (cf. Fig. 2c, middle column).

Then, after a few scans, one oxygen is removed and after some intermediate (not clearly observed) steps, a structure with two defects, a MV and a 555–777 DV, is formed. These observations indicate that, similar to all-carbon defects in graphene, also carbon–oxygen configurations can undergo beam-induced bond rotations and thereby migrate in the lattice[22]. Figure 4g shows the 5-8-5 DV in two distinct, but equivalent states.

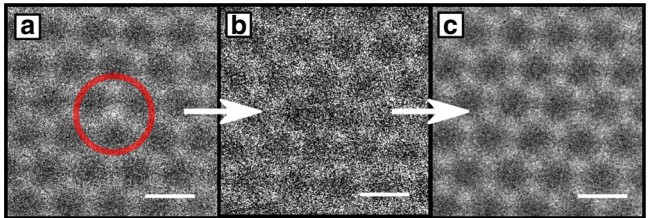

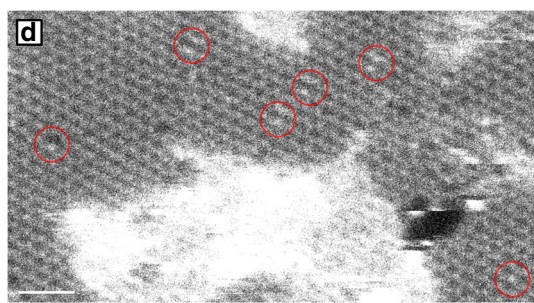

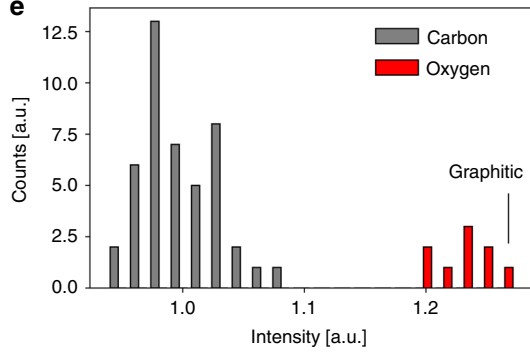

**Fig. 3** Graphitic oxygen substitution. **a** Unprocessed STEM image of a graphitic oxygen substitution in graphene. **b** Oxygen atom is sputtered after four frames, leaving a vacancy. **c** Pristine graphene lattice after the vacancy gets refilled by a carbon atom. **d** Low-magnification image of the GO sample, where multiple pair configurations and a graphitic substitution is present. **e** Histogram of intensity distribution of atoms in panel **d**. Scale bars in panel **a–c** and in panel **d** are 0.25 and 1 nm, respectively

## Discussion

Oxygen with three carbon neighbors appears as a surprise, because it seems to contradict the textbook concept of oxygen forming two bonds (or one double bond), while nitrogen forms three, and carbon up to four covalent bonds. Within the known organic compounds, trivalent oxygen only appears in a charged state, referred to as oxonium, and is difficult to stabilize in extended compounds[28]. Here, the oxygen with three carbon neighbors is found in an extended organic matrix, and the fact that it survives sufficient dose of high-energy electrons for recording several high-resolution images means that it must have a remarkable stability.

With respect to the structure of GO, our results indicate that oxygen atoms incorporated into the graphene lattice or integrated into small defects within in the graphene plane could play an important role among the structural configurations in GO or reduced GO. In particular, the high stability of these configurations means that they would be difficult to remove, e.g. by thermal annealing. Indeed, several of our configurations appear to have been predicted by simulations of oxidation and annealing of graphene (Fig. 2 of ref. [29]), and formed under simulated annealing conditions where most functional groups attached to the basal plane were removed.

In conclusion, we have shown a large variety of bonding configurations of nitrogen and oxygen atoms in a carbon matrix

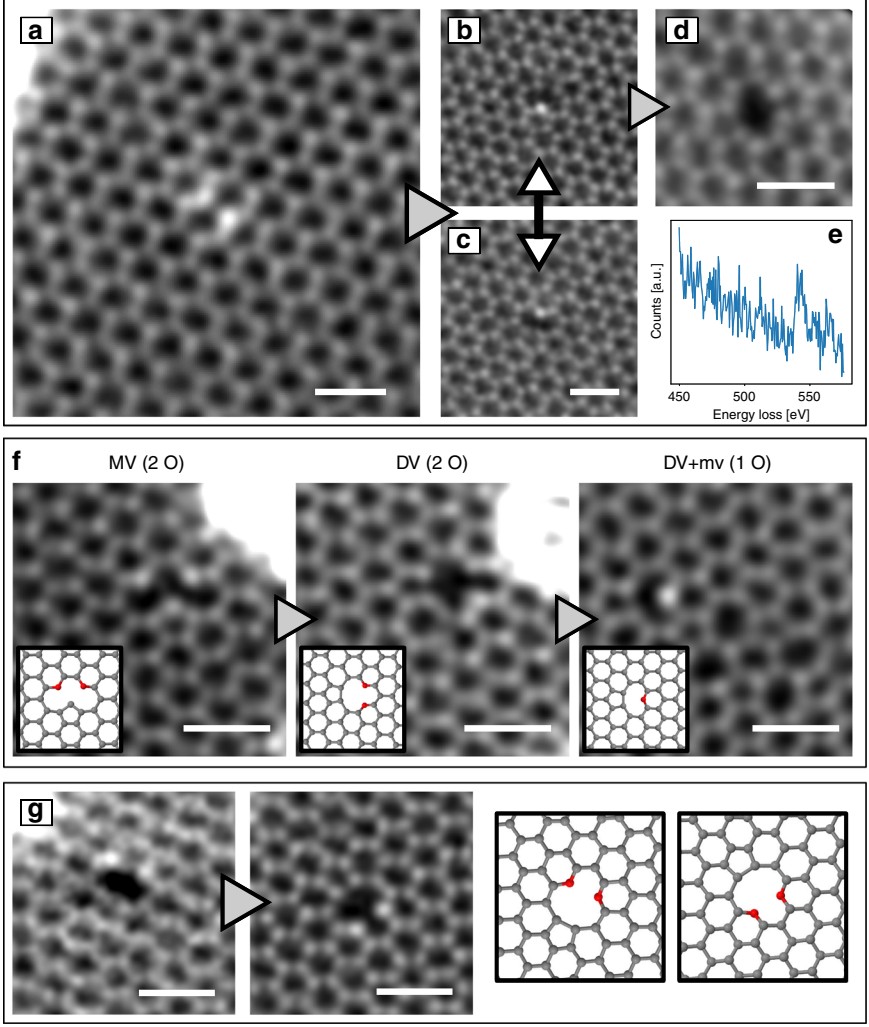

**Fig. 4** In situ oxygen reduction and dynamics in GO. **a** STEM image of the oxygen pair configuration. **b, c** One oxygen is released after several scans, creating a vacancy beside the oxygen atom. The oxygen atom jumps frequently to the opposite equivalent site. **d** Second oxygen is knocked out after several scans creating a divacancy. **e** EEL spectrum of a double-O configuration, which converted into single-O during spectrum acquisition (total dose: ca. $2 \times 10^{10}$ e$^-$ Å$^{-1}$). **f** Reduction process of oxygen. **g** Rotation of the 5-8-5 DV with two oxygen atoms. Scale bar is 0.5 nm

via atomic resolution imaging. For the first time, individual oxygen impurities were clearly identified, and a statistical analysis for both oxygen-containing and nitrogen-containing defects was presented. By and large, the preference of nitrogen for three bonds, versus the preference of oxygen for two bonds, is confirmed. An oxygen pair configuration is revealed to be a very frequent configuration in GO, followed by different types of double-coordinated oxygen atoms at the edges of vacancies. As a remarkable minority, symmetric, graphitic substitutions of oxygen binding to three carbon neighbors in graphene were found. Further, we presented electron-beam-induced reduction dynamics. Overall, we find that the structural features of the defects are similar for all-carbon defects compared to nitrogen-containing or oxygen-containing defects in the same configuration, while differences in the bond lengths or stability are nevertheless detectable.

## Methods

**Sample preparation**. GO is usually prepared from graphite oxide[30–32] by mixing graphite powder into an acid solution, which leads to oxidation. In an improved method, the temperature during oxidation is kept low in order to suppress the extensive formation of $CO_2$, which therefore improves the quality of the sample[33]. Water dispersion of graphene oxide was received from the company Danubia

NanoTech, Ltd. The oxidation method of graphitic powder and subsequent exfoliation were developed with a goal to preserve the long-range structural order in the graphene oxide flakes exfoliated down to the single-atom thickness. Water dispersion of GO was significantly diluted (ca. 1:100). A TEM grid was then vertically dipped into the dispersion for one minute and dried in air afterwards.

Nitrogen-doped samples prepared for comparison were made by irradiating graphene on TEM grids (obtained from Graphenea) with 50 eV nitrogen ions[26]. The plasma irradiation was carried out in a target chamber that is directly connected to the Nion microscope via a UHV transfer system[34]. The sample was irradiated for 16 min at a pressure of ca. $3 \times 10^{-6}$ mbar, resulting in a total ion dose of 4 ions nm$^{-2}$. During irradiation, the sample was heated with a laser (270 mW) in order to reduce contamination. The irradiation treatment in the vacuum system is very similar to the preparation in ref. [34] except that we used nitrogen instead of argon. As a result, we find numerous defects where open bonds can still be observed, e.g., the pyridinic nitrogen configuration (which would likely be covered with contamination if the sample were transferred through air) without post-annealing the sample[5].

**Electron microscopy**. STEM experiments were conducted using a Nion Ultra-STEM100, operated at 60 kV. Typically, our atomic-resolution images were recorded with $512 \times 512$ pixels for a field of view of 6–8 nm and dwell time of 16 μm per pixel using the medium angle annular dark field (MAADF) detector with an angular range of 60–200 mrad. The probe current was ~20 pA and the beam (semi-)convergence angle was 30 mrad. Where appropriate if the structure did not change, a few (2–5) experimental images were averaged in order to increase the signal-to-noise ratio.

**Intensity analysis**. The histograms show integrated atom intensities of the double-Gaussian processed STEM images. The parameters for the filter are similar as in ref. [2] so that the maximum of the double-Gaussian function is between the first two orders of the graphene peaks in the reciprocal space. The width of the Gaussian fit of the carbon peak is assumed to be the same as for the intensity distribution of the other elements.

**Density functional theory**. We used DFT as implemented in the Vienna ab initio simulation package (VASP)[35] within the generalized gradient approximation of Perdew, Burke, and Ernzernhof (PBE) for exchange and correlation[36]. Projector-augmented wave (PAW) potentials[37] were used to describe the core electrons. The kinetic energy cutoff was 700 eV. In case of oxygen impurities, a spin-polarized density-functional method was used. Depending on the defect size and number of impurity atoms, different supercells were selected for modeling. The smallest one was a $5 \times 5 \times 1$ supercell with 50 atoms for the nitrogen graphitic impurity,whereas a $8 \times 8 \times 1$ supercell containing 128 atoms was used for the TV defect. For all defect structures, a $\Gamma$-centered $k$-point sampling was used for the Brillouin-zone integration. The $k$-point meshes were selected to correspond to $36 \times 36 \times 1$ points for the unit cell of graphene. The structures were fully optimized using the damped molecular dynamics method until the residual forces were smaller than 0.005 eV Å$^{-1}$. Due to the size of the defects and existing impurities, special care was devoted to minimize the external pressure or strain on the supercells calculated from traces of the stress tensor. The total energies were calculated based on supercell sizes with minimum external pressure.

To calculate the displacement threshold energy, we carred out DFT/MD calculations at 300 K using the Langvin NVT thermostat. In these simulations, appropriate velocity is given to the oxygen atom and the simulation is run with a 0.5 fs time step for 100 fs. The calculation is then repeated for the neighboring carbon atom.

## Data availability

The full STEM data on which the statistical analysis of the configurations are based are available on figshare with identifier https://doi.org/10.6084/m9.figshare.9205367.v3[38]. Although the data is classified in different sub-folders, each image might contain multiple configurations.

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

## Acknowledgements

This work was supported by the European Research Council (ERC) Grant No. 336453-PICOMAT. M.R.A.M. and J.K. acknowledge support from the Austrian Science Fund (FWF) through project P31605-N36, M.T. and T.S. through project P28322-N36 and V.S. through project no. I2344-N36 and also acknowledges the Slovak Science and Development Funding Agency (APVV) project APVV-16-0319, and T.S. also the European Research Council (ERC) Grant no. 756277-ATMEN.

## Author contributions

C.H. performed the STEM experiments, analysis of GO, and drafted the figures and the manuscript. V.S. prepared GO samples and participated in STEM measurements. T.G., A.M. and M.T. performed the synthesis, STEM experiments, and analysis of N-doped graphene. M.R.A.M. and T.S. performed the DFT simulations. C.M. prepared alignments for the STEM instrument and supported the experiments. J.K. and J.C.M conceived and supervised the study.

## Competing interests

The authors declare no competing interests.
