## [Peer Review File · Nature Communications]

Reviewers' Comments:

Reviewer #1:

Remarks to the Author:

The authors report on the STEM direct imaging of oxygen and nitrogen impurity atoms in (reduced) graphene oxide. The experiments constitute a tour de force and deserve to be publication. However, although very interesting, some findings would need some additional confirmation before publication.

The main original finding of the work is the observation of triple-coordinated oxygen atoms with three carbon atoms from the graphene network. As mentioned by the authors, such a bonding is highly unusual in organic compounds.

To draw this conclusion, the authors compare STEM HAADF images to computed relaxed structures.

1) The main question here concerns the dynamics of the observation. How can the authors be sure that the observed conformations do not correspond to out of equilibrium states?

Some sequences have been recorded for N atoms jumping between 2 bonding configurations. The dose per image was $6.105 \text{ e}^- \text{ \AA}^{-2}$ per image which is still a significant dose.

Did the authors tried similar acquisitions at lower dose for oxygen atoms? If feasible, this could provide some answer to the question of the stability of such triple-coordinated oxygen bonds.

2) The authors exclude the presence of functional groups (bound to be destroyed at dose between 103 and $106 \text{ e}^-/\text{\AA}$). In connection to the previous comment, it would be interesting to show images at $103 \text{ e}^-/\text{\AA}$, if feasible.

3) How was the EELS done at such a low dose?

4) How many images/configuration were acquired? Giving a number would be important as the authors mention a statistical analysis of the configurations distribution.

Reviewer #2:

Remarks to the Author:

This is a very interesting manuscript that describes the imaging of the light elements oxygen (O) and nitrogen (N) in as-prepared graphene oxide samples prepared in-house and nitrogen-sputtered graphene specimens also prepared in house. They have used an annular dark-field Scanning Transmission Electron Microscopy (i.e. ADF-STEM) imaging approach (reference 2) which is demonstrably more sensitive to distinguishing between these lighter elements than other aberration corrected electron imaging techniques. Their results could be of wide interest to other researchers in this field but I suggest that they address the following points at least to the satisfaction of the editor if not by a second review before this manuscript should be passed for publication in Nature Communications (Note: some corrections are typographical or presentational only but points (IV) and (VII) do require particular attention):

(I) Page 2, 6 lines down – perhaps replace “...allows to distinguish...” with “...allows the distinction...”

(II) Page 3 Figure 1 – the insets in these figures are much too small to the extent that the functional groups in the inset in panel a are poorly visible on the printed page. Similarly, the legend in panel c is barely legible. Can the authors reconfigure this Figure to make these items more distinguishable ?

(III) Page 4, end of first paragraph – insert comma after EELS, i.e. “These structures, which are stable enough for STEM imaging and in some cases EELS reveal a variety of...” should become “These structures, which are stable enough for STEM imaging and in some cases EELS, reveal a variety of...”

(IV) Page 6 Question regarding the paragraph “Oxygen with three carbon neighbors appears as a surprise, because it seems to contradict the textbook concept of oxygen forming two bonds (or one double bond), while nitrogen forms three, and carbon up to four covalent bonds. Within the known organic compounds, trivalent oxygen only appears in a charged state, referred to as oxonium, and is difficult to stabilize in extended compounds [24]. Here, the oxygen with three carbon neighbors is found in an extended organic matrix, and the fact that it survives sufficient dose of high-energy electrons for recording several high-resolution images means that it must have a remarkable stability.” This is possibly controversial and cannot pass without comment. In the next paragraph, the authors discuss the use of DFT methods to establish the energy thresholds for removing oxygen or carbon atoms from a 2D graphene lattice – can they use a similar approach to establish whether or not the formation energy of an Oxygen atom with three nearest neighbours as they describe is energetically favourable or does theory optimize a different configuration ?

(V) Page 7, Figure 2e and caption. Can the authors indicate more clearly the size of the sample used to establish the data in this figure ?

(VI) Page 8 Figure 3, panels a,b,c – the authors indicate that firstly an oxygen atom in a graphene lattice is sputtered (panel a), which is clearly visible, leaving a vacancy (panel b), which is only very weakly visible, followed by filling with a carbon atom (panel c) which is much more clearly visible. Can the authors improve the contrast balance between these three panels such that the sequence they describe is consistently visible across all three images ?

(VII) A general comment and question: the authors describe the method that was used to prepare the graphene oxide used in their experiment (i.e. Methods section) but also point out, in their introduction, the well-known controversy about the local structure of this material. Are the authors able to comment at all about their results vis-à-vis graphene oxide prepared by other methods, making use if necessary of the extended literature ?

Reviewers' comments:

Reviewer #1 (Remarks to the Author):

The authors report on the STEM direct imaging of oxygen and nitrogen impurity atoms in (reduced) graphene oxide. The experiments constitute a tour de force and deserve to be publication. However, although very interesting, some findings would need some additional confirmation before publication.

The main original finding of the work is the observation of triple-coordinated oxygen atoms with three carbon atoms from the graphene network. As mentioned by the authors, such a bonding is highly unusual in organic compounds.

To draw this conclusion, the authors compare STEM HAADF images to computed relaxed structures.

1) The main question here concerns the dynamics of the observation. How can the authors be sure that the observed conformations do not correspond to out of equilibrium states?

Some sequences have been recorded for N atoms jumping between 2 bonding configurations. The dose per image was $6.105 \text{ e}^- \text{ \AA}^{-2}$ per image which is still a significant dose.

Did the authors tried similar acquisitions at lower dose for oxygen atoms? If feasible, this could provide some answer to the question of the stability of such triple-coordinated oxygen bonds.

We thank the reviewer for the encouraging comments.

For the experiment, we have chosen a minimum dose where it is still possible to retrieve enough signal for the distinction of oxygen and carbon. It is already one order of magnitude lower than the dose used in the initial work that demonstrated the identification of light elements from the ADF intensity (cf. Krivanek et al. *Nature*, volume 464, pages 571–574, 2010). Further reduction would suppress the signal too much for proper analysis of the structure and chemical identity.

The stability can be confirmed by the absence of line-to-line variations in the raw STEM data: If an atomic configuration changes between subsequent scan-lines, it appears like atoms split into segments.

The example image shows a graphene edge, where the configuration clearly has changed between some subsequent scan lines:

In contrast, oxygen atoms appear continuous in the raw STEM data:

The line-to-line variations also reveals an oxygen jump within one frame.

The first frame is a static configuration, in the second frame, the atom jumps between the vacancy sites (indicated by the change in the subsequent scan-lines) and the third configuration is static again.

In addition, the triple-coordinated oxygen configurations lasted several scans before a beam-induced configuration change, which further confirms their stability.

2) *The authors exclude the presence of functional groups (bound to be destroyed at dose between 103 and 106 e-/Å). In connection to the previous comment, it would be interesting to show images at 103 e-/Å, if feasible.*

Resolving single atoms (as shown in the Figures of the main manuscript) required a dose of at least several 10^6 e-/Å². An image with a significant lower dose (e.g. 10^3 e-/Å) is dominated by noise. Unfortunately, the limited beam stability makes it impossible to directly image individual functional groups.

3) *How was the EELS done at such a low dose?*

For the dose-dependent EELS in Fig. 1c+d, we zoomed out to a field of view of 16 nm. This larger area results in a dose of ca. 5000 e-/Å² per acquisition. For the visibility of the oxygen edge intensities in the EELS, we averaged 4 spectra resulting in a dose of $2 \cdot 10^4$ e-/Å² per data point. Due to the lower dose, we can assume that oxygen-containing functional groups are still present at the beginning, but inevitably the spectra are averaged over many atoms.

The EELS in Fig. 4 in contrast was obtained from an individual impurity, and at much higher dose. In that case, the atomic configuration did change, but nevertheless the chemical identity could be confirmed. The total dose is now given in the figure caption:

“(e) EEL spectrum of another double-O configuration, which converted into single-O during spectrum acquisition (total dose: ca. $2 \cdot 10^{10}$ e-/Å²)”

4) *How many images/configuration were acquired? Giving a number would be important as the authors mention a statistical analysis of the configurations distribution.*

We included the number of samples for both elements in the manuscript (Fig. 2).

Reviewer #2 (Remarks to the Author):

This is a very interesting manuscript that describes the imaging of the light elements oxygen (O) and nitrogen (N) in as-prepared graphene oxide samples prepared in-house and nitrogen-sputtered graphene specimens also prepared in house. They have used an annular dark-field Scanning Transmission Electron Microscopy (i.e. ADF-STEM) imaging approach (reference 2) which is demonstrably more sensitive to distinguishing between these lighter elements than other aberration corrected electron imaging techniques. There results could be of wide interest to other researchers in this field but I suggest that they address the following points at least to the satisfaction of the editor if not by a second review before this manuscript should be passed for publication in Nature Communications (Note: some corrections are typographical or presentational only but points (IV) and (VII) do require particular attention):

(I) Page 2, 6 lines down – perhaps replace “...allows to distinguish...” with “...allows the distinction...”

(II) Page 3 Figure 1 – the insets in these figures are much too small to the extent that the functional groups in the inset in panel a are poorly visible on the printed page. Similarly, the legend in panel c is barely legible. Can the authors reconfigure this Figure to make these items more distinguishable ?

(III) Page 4, end of first paragraph – insert comma after EELS, i.e. “These structures, which are stable enough for STEM imaging and in some cases EELS reveal a variety of...” should become “These structures, which are stable enough for STEM imaging and in some cases EELS, reveal a variety of...”

We thank the reviewer for the encouraging comments. We modified the text and the Figure as the reviewer suggested. For point (I) we used as slightly different phrasing.

(IV) Page 6 Question regarding the paragraph “Oxygen with three carbon neighbors appears as a surprise, because it seems to contradict the textbook concept of oxygen forming two bonds (or one double bond), while nitrogen forms three, and carbon up to four covalent bonds. Within the known organic compounds, trivalent oxygen only appears in a charged state, referred to as oxonium, and is difficult to stabilize in extended compounds [24]. Here, the oxygen with three carbon neighbors is found in an extended organic matrix, and the fact that it survives sufficient dose of high-energy electrons for recording several high-resolution images means that it must have a remarkable stability.” This is possibly controversial and cannot pass without comment. In the next paragraph, the authors discuss the use of DFT methods to establish the energy thresholds for removing oxygen or carbon atoms from a 2D graphene lattice – can they use a similar approach to establish whether or not the formation energy of an Oxygen atom with three nearest neighbours as they describe is energetically favourable or does theory optimize a different configuration ?

The DFT values are now included as a table in the supplementary information, and the following text is included in the manuscript:

We used DFT to calculate the energy of the relaxed configurations with the heteroatoms incorporated in the lattice for both, nitrogen and oxygen (E_{in}) (see supplementary information). We also calculate the sum of the energies ($E_{out}+E_{isolated}$) when the heteroatoms are released from the lattice (after relaxation) and the isolated atoms (the half of N_2 or O_2). The difference $E_{in}-(E_{out}+E_{isolated})$ is referred as binding

energy and is lower (meaning higher stability) for all N configurations which is expected due to the higher stability of N dopants in graphene. The binding energies are negative in all cases, which means that the structures are stable with respect to forming a carbon-only vacancy plus isolated O or N.

(V) Page 7, Figure 2e and caption. Can the authors indicate more clearly the size of the sample used to establish the data in this figure ?

We included the number of samples for both elements in the manuscript.

(VI) Page 8 Figure 3, panels a,b,c – the authors indicate that firstly an oxygen atom in a graphene lattice is sputtered (panel a), which is clearly visible, leaving a vacancy (panel b), which is only very weakly visible, followed by filling with a carbon atom (panel c) which is much more clearly visible. Can the authors improve the contrast balance between these three panels such that the sequence they describe is consistently visible across all three images ?

The different quality of the image arises from the different dose used and therefore different signal-to-noise ratio of the frames. The first image is an average of 4 frames, the second one is a single frame and the third is an average of 10 frames (as described in methods, we averaged frames when the configuration has not changed).

Therefore, the images inevitably have a different signal to noise ratio, and hence can not be made to appear in the same way. Nevertheless we have tried to improve the contrast range.

In addition, we deposited all of the raw data onto a public server (reference), and give the reference in the manuscript.

(VII) A general comment and question: the authors describe the method that was used to prepare the graphene oxide used in their experiment (i.e. Methods section) but also point out, in their introduction, the well-known controversy about the local structure of this material. Are the authors able to comment at all about their results vis-à-vis graphine oxide prepared by other methods, making use if necessary of the extended literature ?

We don't assume that the observed configurations are specific to our samples, and hence we wouldn't want to make comparisons of this sample to GO from other works. The configurations may have gone unnoticed in earlier HRTEM studies due to the nearly identical contrast of C, N and O in HRTEM images.

However, we now briefly discuss the observations in the context of GO configurations and in particular compare our observation with a simulated study of annealed (reduced) GO. We added a short paragraph to the paper (just before conclusions):

“With respect to the structure of GO, our results indicate that oxygen atoms incorporated into the graphene lattice or integrated into small defects within the graphene plane could play an important role among the structural configurations in GO or reduced GO. In particular, the high stability of these configurations means that they would be difficult to remove e.g. by thermal annealing. Indeed, several of our configurations, including the double-oxygen or graphitic oxygen, appear to have been predicted by simulations of oxidation and annealing of graphene (Fig. 2 of [

<https://www.nature.com/articles/ncomms9335>), and formed under simulated annealing conditions where most functional groups attached to the basal plane were removed.”

Reviewers' Comments:

Reviewer #1:

Remarks to the Author:

The Authors have addressed all my concerns. The paper is now suitable for publication.

Reviewer #2:

Remarks to the Author:

In general I find that this manuscript is much improved and that the authors have been at pains to address all of the comments and queries posed by both referees. I do have some final additional comments and one question which should possibly address to the satisfaction of the editor before this article is passed for publication in Nature Communications.

The additional minor corrections are recommended (no further review needed):

(I)' Page 4, Figure 1 caption. Change "High magnification double-gaussian..." to "High magnification double-Gaussian..."

(II)' Page 8 - I still find Figs 3a-c confusing (i.e. my original comment (VI)). This might look better if the authors present this sequence of images as a three image series plus one enlargement i.e. Panel a (enlargement) then panels a->b->c in a sequence with identical regions and enlargements being presented. The evolution of this region as a function of beam irradiation should then become obvious. QUESTION: Secondly, can the authors rule out that the bright atom in Figure 3a (as presented) is not a C-O pair (i.e. as in a hydroxyl \geq C-O-H functionality, often proposed GO and which includes tetrahedral coordination for carbon, for example) ?

(III)' Page 12 'Intensity Analysis' section - change all "double-gaussian" or "gaussian" to "double-Gaussian" or "Gaussian", respectively (see comment (I' above) as Gaussian is derived from a proper name.

Reviewers' comments:

Reviewer #1 (Remarks to the Author):

The Authors have addressed all my concerns. The paper is now suitable for publication.

We thank the reviewers contribution to the improvement of the work.

Reviewer #2 (Remarks to the Author):

In general I find that this manuscript is much improved and that the authors have been at pains to address all of the comments and queries posed by both referees. I do have some final additional comments and one question which should possibly address to the satisfaction of the editor before this article is passed for publication in Nature Communications.

The additional minor corrections are recommended (no further review needed):

(I)' Page 4, Figure 1 caption. Change "High magnification double-gaussian..." to "High magnification double-Gaussian..."

(II)' Page 8 - I still find Figs 3a-c confusing (i.e. my original comment (VI)). This might look better if the authors present this sequence of images as a three image series plus one enlargement i.e. Panel a (enlargement) then panels a->b->c in a sequence with identical regions and enlargements being presented. The evolution of this region as a function of beam irradiation should then become obvious. QUESTION: Secondly, can the authors rule out that the bright atom in Figure 3a (as presented) is not a C-O pair (i.e. as in a hydroxyl \geq C-O-H functionality, often proposed GO and which includes tetrahedral coordination for carbon, for example) ?

(III)' Page 12 'Intensity Analysis' section - change all "double-gaussian" or "gaussian" to "double-Gaussian" or "Gaussian", respectively (see comment (I' above) as Gaussian is derived from a proper name.

We thank the reviewers contribution to the improvement of the work. We applied all of the suggested changes in the manuscript. We now present Fig. 3a-c series of three images, at a size where no enlargement is needed. We can rule out that the bright atom in Fig. 3a is a C-O pair, because then the two atoms behind each other in projection would lead to a much higher intensity.